# Towards Robust Bisimulation Metric Learning

**Mete Kemertas**[*]
Department of Computer Science
University of Toronto
kemertas@cs.toronto.edu

**Tristan Aumentado-Armstrong**[*]
Department of Computer Science
University of Toronto
taumen@cs.toronto.edu

## Abstract

Learned representations in deep reinforcement learning (DRL) have to extract task-relevant information from complex observations, balancing between robustness to distraction and informativeness to the policy. Such stable and rich representations, often learned via modern function approximation techniques, can enable practical application of the policy improvement theorem, even in high-dimensional continuous state-action spaces. Bisimulation metrics offer one solution to this representation learning problem, by collapsing functionally similar states together in representation space, which promotes invariance to noise and distractors. In this work, we generalize value function approximation bounds for on-policy bisimulation metrics to non-optimal policies and approximate environment dynamics. Our theoretical results help us identify embedding pathologies that may occur in practical use. In particular, we find that these issues stem from an underconstrained dynamics model and an unstable dependence of the embedding norm on the reward signal in environments with sparse rewards. Further, we propose a set of practical remedies: (i) a norm constraint on the representation space, and (ii) an extension of prior approaches with intrinsic rewards and latent space regularization. Finally, we provide evidence that the resulting method is not only more robust to sparse reward functions, but also able to solve challenging continuous control tasks with observational distractions, where prior methods fail.

## 1 Introduction

Complex reinforcement learning (RL) problems require the agent to infer a useful representation of the world state from observations. The utility of this representation can be measured by how readily it can be used to learn and enact a policy that solves a specific task. As an example, consider a robot using visual perception to pour a cup of coffee. The clutter on the counter and the colour of the walls have little effect on the correct action; even more pertinent aspects, such as two mugs with slightly different patterns and shapes, should be treated nearly the same by the policy. In other words, many states are *equivalent* for a given task, with their differences being task-irrelevant distractors. Thus, a natural approach to generalization across environmental changes is constructing a representation that is invariant to such nuisance variables – effectively "grouping" such equivalent states together.

One method of obtaining such a representation uses the notion of a *bisimulation metric* (BSM) [13, 14]. The goal is to abstract the states into a new metric space, which groups "behaviourally" similar states in a task-dependent manner. In particular, bisimilar states (i.e., in this case, those close under the BSM) should yield similar stochastic reward sequences, given the same action sequence. In a recursive sense, this requires bisimilar states to have similar (i) immediate rewards and (ii) transition distributions in the BSM space (e.g., see Eq. 1). Thus, ideally, the resulting representation space contains the information necessary to help the policy maximize return and little else.

---

[*]Equal contribution.

35th Conference on Neural Information Processing Systems (NeurIPS 2021).

Recently, the Deep Bisimulation for Control (DBC) algorithm [53] showed how to map observations into a learned space that follows a policy-dependent (or on-policy) BSM (PBSM) for a given task [10], resulting in a powerful ability to ignore distractors within high-dimensional observation spaces. Using a learned dynamics model, DBC trains an encoder to abstract states to follow the estimated PBSM, using the aforementioned recursive formulation for metric learning.

However, this method can suffer from issues of robustness under certain circumstances. Conceptually, differentiating state encodings requires trajectories with different rewards. In other words, two states that only differ in rewards far in the future should still not be bisimilar. Yet, in practice, the encoder and policy are learned simultaneously as the agent explores; hence, in the case of *uninformative* (e.g., sparse or near-constant) rewards, it may incorrectly surmise that two states are bisimilar, as most trajectories will look very similar. This leads to a premature *collapse* of the embedding space, grouping states that should not be grouped. On the other hand, we can show that the formulation of the metric learning loss is susceptible to embedding *explosion* if the representation space is left unconstrained[2]. In our work, we build upon the DBC model in an attempt to tackle both problems: (1) we address embedding explosion by stabilizing the state representation space via a norm constraint and (2) we prevent embedding collapse by altering the encoder training method.

**Contributions** We (i) generalize theoretical value function approximation bounds to a broader family of BSMs, with learned dynamics; (ii) analyze the BSM loss formulation and identify sources of potential embedding pathologies (explosion and collapse); (iii) show how to constrain representation space to obtain better embedding quality guarantees; (iv) devise theoretically-motivated training modifications based on intrinsic motivation and inverse dynamics regularization; and (v) show on a variety of tasks that our method is more robust to both sparsity and distractors in the environment.

## 2 Related Work

Our work builds on two major aspects of DRL: representation learning, particularly task-specific encodings following the BSM, and methods for sparse reward environments, (e.g., intrinsic motivation).

**Representation Learning for Control** In the context of RL, there is a spectrum of representations from task-agnostic to task-specialized. One task-agnostic approach to learning latent representation spaces uses a reconstruction loss. In particular, some methods utilize this approach for model-based planning [27, 54, 25], while others do so for model-free RL [33, 17, 30]. Such approaches often utilize generative models, which have shown use for learning via simulation, in addition to representation learning [22, 51]. Instead of fully reconstructing an input, other approaches use self-supervised representation learning methods, especially contrastive learning algorithms [34, 28, 44].

While task-agnostic reconstruction is reusable and transferable, it can store unnecessary or distracting details. In contrast, *state abstraction methods* [31] can obtain compact state representations that ignore irrelevant information, by finding (approximately) equivalent aggregated MDPs that have (nearly) identical optimal policies. To learn such MDP homomorphisms [38, 39], previous efforts have exploited various structural regularities (e.g., equal rewards and transition probabilities [19, 46], state-action symmetries and equivariances [46, 48, 49]) and assumptions [6]. *State similarity metric learning* [29] is closely related, as one can aggregate $\epsilon$-close states to produce a new MDP. For example, inter-state distances can be assigned based on the difference between policy distributions conditioned on respective states [2, 18]. Inverse dynamics prediction can also help encourage holding only information relevant to what the agent can control [3, 36]. For model-based RL, prior work constructed equivalence classes of *models* [21], as well as other abstractions [12, 16], to save computation and memory.

In our work, we build on the use of task-specific *bisimulation relations* [19], which define equivalence classes of states based on rewards and transition probabilities. However, obtaining equivalence classes is brittle, as our estimates of these quantities may be inexact. Hence, Ferns et al. [13, 14, 15] instead turn to a bisimulation *metric* between states, which can vary smoothly as the rewards and transition probabilities change. More recently, Zhang et al. [53] devised Deep Bisimulation for Control (DBC), which applies a metric learning approach to enforce its representation to approximately follow bisimulation-derived state aggregation. While DeepMDP [17] proves that its representation provides an upper bound on the bisimulation distance, DBC [53] directly imposes the bisimulation metric

---

[2]Embedding collapse and explosion issues also appear in the computer vision literature [26, 42, 52].

structure on the latent space. Herein, we extend prior analysis and methods for representation learning using PBSMs, with the goal of improving robustness and general performance. In particular, we focus on preventing embedding pathologies, in the context of uninformative reward signals and distraction.

**Reinforcement Learning under Uninformative Rewards** Many real-world tasks suffer from un-informative rewards, meaning that the agent receives sparse (mostly zero) or static (largely constant) rewards throughout its trajectories. For example, games like Montezuma's Revenge only provide reward signals after a large number of steps – and the agent is likely not to receive one at all, in most cases. Thus, while common, this situation significantly increases the difficulty of the RL task, as the agent receives little signal about its progress. To deal with such situations, a common tactic is to encourage the agent to explore in the absence of an extrinsic reward signal. This is often done by providing *intrinsic motivation* via a self-derived reward, resulting in "curiosity-driven" behaviour [8, 4]. Such approaches include surprise [1, 5] (where experiencing unexpected dynamics is rewarded), and empowerment [40, 20] (where the agent prefers states in which it has more control).

In this work, we specifically target the case of uninformative rewards, showing that the DBC model is especially susceptible to instability or collapse for such tasks, and consider ways to ameliorate this issue. In particular, we utilize the forward prediction error as an intrinsic reward [36, 8, 43], thus augmenting the sparse extrinsic rewards and encouraging exploration. We also regularize the latent space by learning an inverse dynamics model on top of it, which does not rely on the extrinsic reward signal and has previously been used for distraction-robust representation learning in DRL [3, 36].

## 3 Technical Approach

In this section, we first describe our notation and problem setting, and review prior definitions of bisimulation metrics and relevant metric learning objectives. Then, we consider theoretical connections to value functions and convergence properties, and show that these formulations may be susceptible to (i) instabilities in optimization due to embedding explosion (i.e., large norms), and (ii) convergence to trivial solutions where all states are represented as a single point (embedding collapse). Based on our analysis, we propose an extension of deep bisimulation metric learning, with theoretically-motivated constraints on the optimization objective, including alterations to the forward dynamics model, intrinsic motivation, and inverse dynamics regularization.

### 3.1 Preliminaries

In this work, we consider a discounted Markov Decision Process (MDP) given by a tuple, $\langle \mathcal{S}, \mathcal{A}, \mathcal{P}, R, \rho_0 \rangle$. At the beginning of each episode, an initial state, $\mathbf{s}_0 \in \mathcal{S}$, is sampled from the initial-state distribution $\rho_0$ over the state space $\mathcal{S}$. Then, at each discrete time-step $t \geq 0$, an agent takes an action, $\mathbf{a}_t \in \mathcal{A}$, according to a policy $\pi(\mathbf{a}_t|\mathbf{s}_t)$. As a result, the MDP transitions to the next state according to a transition distribution $\mathcal{P}(\mathbf{s}_{t+1}|\mathbf{s}_t, \mathbf{a}_t)$. The agent collects a scalar reward, $r_t = R(\mathbf{s}_t, \mathbf{a}_t)$, from the environment according to a bounded reward function, $R : \mathcal{S} \times \mathcal{A} \to [R_{\min}, R_{\max}]$. In infinite- and long-horizon settings, a discount factor, $\gamma \in [0, 1)$, is used to calculate the agent's discounted return in a given episode, $G = \sum_{t \geq 0} \gamma^t r_t$. RL algorithms aim to find an optimal policy, $\pi^* := \mathrm{argmax}_{\pi \in \Pi} \mathbb{E}[G]$, for a class of stationary policies $\Pi$. In high-dimensional, continuous state (or observation) spaces, this learning problem is rendered tractable via a state encoder (e.g., a neural network), $\phi : \mathcal{S} \to \mathbb{R}^n$, which is used to learn a policy of the form $\pi(\mathbf{a}|\phi(\mathbf{s}))$.

The following (pseudo[3])-metric, based on the Wasserstein metric (see Appendix A for a review), is of particular relevance to this work. Distances are assigned to state pairs, $(\mathbf{s}_i, \mathbf{s}_j) \in \mathcal{S} \times \mathcal{S}$, according to a pessimistic measure [9] of how much the rewards collected in each state and the respective transition distributions differ. A distance of zero for a pair implies state aggregation, or *bisimilarity*.

**Definition 1** (Bisimulation metric for continuous MDPs, Thm. 3.12 of [14])**.** *The following metric exists and is unique, given $R : \mathcal{S} \times \mathcal{A} \to [0, 1]$ and $c \in (0, 1)$ for continuous MDPs:*

$$d(\mathbf{s}_i, \mathbf{s}_j) = \max_{\mathbf{a} \in \mathcal{A}} (1 - c)|R(\mathbf{s}_i, \mathbf{a}) - R(\mathbf{s}_j, \mathbf{a})| \ + \ c W_1(d)(\mathcal{P}(\cdot|\mathbf{s}_i, \mathbf{a}), \mathcal{P}(\cdot|\mathbf{s}_j, \mathbf{a})). \quad (1)$$

An earlier version of this metric for finite MDPs used separate weighting constants $c_R, c_T \geq 0$ for the first and second terms respectively, and required that $c_R + c_T \leq 1$ [13]. Here, when the weighting

---

[3]Pseudo-metrics are a generalization of metrics that allow distinct points to have zero distance. For simplicity, we broadly use the term "metric" in the remainder of this paper at the cost of imprecision.

constant $c_T$ of the $W_1(d)$ term is in $[0, 1)$, the RHS is a contraction mapping, $\mathcal{F}(d) : \mathfrak{met} \to \mathfrak{met}$, in the space of metrics. Then, the Banach fixed-point theorem can be applied to ensure the existence of a unique metric, which also ensures convergence via fixed-point iteration for finite MDPs. For more details, we refer the reader to [13, 14] and the proof of Remark 1 in Appendix B. Notice that $c$ (or $c_T$) determines a timescale for the bisimulation metric, weighting the importance of current versus future rewards, analogously to the discount factor $\gamma$. More recently, an *on-policy* bisimulation metric (also called $\pi$-bisimulation) was proposed to circumvent the intractibility introduced by taking the $\max$ operation over high-dimensional action spaces (e.g., continuous control), as well as the inherent pessimism of the policy-independent form [10].

**Definition 2** (On-policy bisimulation metric [10]). *Given a fixed policy $\pi$, the following on-policy bisimulation metric exists and is unique:*

$$d_\pi(\mathbf{s}_i, \mathbf{s}_j) := |r_i^\pi - r_j^\pi| \; + \; \gamma W_1(d_\pi)(\mathcal{P}^\pi(\cdot|\mathbf{s}_i), \mathcal{P}^\pi(\cdot|\mathbf{s}_j)), \tag{2}$$

*where $r_i^\pi := \mathbb{E}_{\mathbf{a}\sim\pi}[R(\mathbf{s}_i, \mathbf{a})]$ and $\mathcal{P}^\pi(\cdot|\mathbf{s}_i) := \mathbb{E}_{\mathbf{a}\sim\pi}[\mathcal{P}(\cdot|\mathbf{s}_i, \mathbf{a})]$.*

Zhang et al. [53] proposed to learn a similar on-policy bisimulation metric directly in the embedding space via an MSE objective. They proposed an algorithm for jointly learning a policy $\pi(\mathbf{a}|\phi(\mathbf{s}))$ with an on-policy bisimulation metric. Below, we define a generalized variant of their objective:

$$J(\phi) := \frac{1}{2}\mathbb{E}\left[\left(\widehat{d}_{\pi,\phi}(\mathbf{s}_i, \mathbf{s}_j) - |r_i^\pi - r_j^\pi| \; - \; \gamma W_2(\|\cdot\|_{q_1})(\widehat{\mathcal{P}}^\pi(\cdot|\phi(\mathbf{s}_i)), \widehat{\mathcal{P}}^\pi(\cdot|\phi(\mathbf{s}_j)))\right)^2\right], \tag{3}$$

where $\widehat{d}_{\pi,\phi}(\mathbf{s}_i, \mathbf{s}_j) := \|\phi(\mathbf{s}_i) - \phi(\mathbf{s}_j)\|_{q_2}$ and they used $(q_1 = 2, q_2 = 1)$. Notice that the recursion induced by this objective is different from prior metrics in three ways; (i) a $W_2$ metric was used instead of a $W_1$ metric since $W_2$ has a convenient closed form for Gaussian distributions when $q_1 = 2$, (ii) the distance function used for the bisimulation metric and the Wasserstein metric are different ($L_1$ and $L_2$ respectively), and (iii) a forward dynamics model is used instead of the ground truth dynamics. While they may introduce practical benefits, these differences violate the conditions under which the existence of a unique bisimulation metric has been proven [10, 13, 14, 15]. Thus, we will (i) assume a unique metric exists for all Wasserstein metrics $W_p$ when necessary (see Assumption 1), (ii) study losses that use a matching metric, i.e., $q_1 = q_2 = q$, and (iii) introduce a constraint on forward models in Sec. 3.2.2.

Next, we note that they recommend the use of stop gradients for the $W_2$ term. The resulting gradient updates may be considered as approximate fixed-point iteration in the space of metrics:

$$\widehat{\mathcal{F}}(\widehat{d}_\pi(\phi_n, \widehat{\mathcal{P}}))(\mathbf{s}_i, \mathbf{s}_j) := \widehat{d}_\pi(\mathbf{s}_i, \mathbf{s}_j; \phi_n + \alpha_n \nabla_\phi J(\phi_n), \widehat{\mathcal{P}}), \tag{4}$$

where $\alpha_n$ is a learning rate. However, in practice, $\pi$ and $\widehat{\mathcal{P}}$ may also be updated in training, which is of particular relevance when they have form $\pi(\mathbf{a}|\phi(\mathbf{s}))$ and $\widehat{\mathcal{P}}(\phi(\mathbf{s}')|\phi(\mathbf{s}), \mathbf{a})$ as in [53]. In the next section, we will discuss conditions under which joint updates to a policy $\pi$ and a metric defined by state encodings $\phi$ may or may not converge in practical settings:

$$\lim_{n\to\infty} \mathcal{F}^{(n)}(\pi, \widehat{d}_{\pi,\phi}) \stackrel{?}{=} d_{\pi^*}, \tag{5}$$

where $d_{\pi^*}$ is the on-policy bisimulation metric for the optimal policy $\pi^*$.

## 3.2 Theoretical Analysis

In this section, we generalize prior value function bounds for bisimulation metrics to cases where arbitrary weighting constants $c_R, c_T$ are used, and further derive a value function approximation (VFA) bound for $V^\pi$ rather than $V^*$ unlike prior work [13, 14, 53]. We then describe constraints on forward dynamics models for convergence to a unique metric in joint training, and derive VFA bounds as a function of forward dynamics modelling error. Then, we discuss potential pitfalls in on-policy bisimulation metric learning, which lead us to connecting its weaknesses with sparse rewards. Our findings motivate our proposed remedies to the issues we identify in on-policy bisimulation; we recommend a particular norm constraint on the latent space, and motivate the use of inverse dynamics and intrinsic motivation outlined in Sec. 3.3. All proofs are provided in Appendix B.

### 3.2.1 Value Function Bounds

An important feature of bisimulation metrics is their relation to value functions since a provably tight connection implies guarantees in VFA. Similarly to previous bounds for policy-independent bisimulation metrics [13, 14], Castro [10] showed that given any two states $(\mathbf{s}_i, \mathbf{s}_j)$, the following bound holds for on-policy bisimulation metrics (see Definition 2): $|V^\pi(\mathbf{s}_i) - V^\pi(\mathbf{s}_j)| \leq d_\pi(\mathbf{s}_i, \mathbf{s}_j)$. As discussed earlier, differently from previous approaches, Zhang et al. [53] used a 2-Wasserstein metric due to practical reasons. Here, in order to generalize previous value function bounds, we assume the existence of a unique bisimulation metric for $p$-Wasserstein metrics.

**Assumption 1** (A1, $p$-Wasserstein bisimulation metric). *For a given $c_T \in [0, 1)$, $c_R \in [0, \infty)$ and $p \geq 1$, the following bisimulation metric exists and is unique:*

$$d_\pi(\mathbf{s}_i, \mathbf{s}_j) := c_R |r_i^\pi - r_j^\pi| + c_T W_p(d_\pi)(\mathcal{P}^\pi(\cdot|\mathbf{s}_i), \mathcal{P}^\pi(\cdot|\mathbf{s}_j)). \tag{6}$$

**Remark 1.** *If $p = 1$, or both the environment and policy are deterministic, A1 holds.*

**Theorem 1** (Generalized value difference bound). *Let the reward function be bounded as $R \in [0, 1]$. For an on-policy bisimulation metric given by Eq. (6), for any $c_T \in [0, 1)$ and $p \geq 1$, define $\overline{\gamma} = \min(c_T, \gamma)$. Given A1, the bisimulation distance between a pair of states upper-bounds the difference in their values:*

$$c_R |V^\pi(\mathbf{s}_i) - V^\pi(\mathbf{s}_j)| \leq d_\pi(\mathbf{s}_i, \mathbf{s}_j) + \frac{2c_R(\gamma - \overline{\gamma})}{(1-\gamma)(1-c_T)}, \ \forall (\mathbf{s}_i, \mathbf{s}_j) \in \mathcal{S} \times \mathcal{S}. \tag{7}$$

Note that Thm. 3 of [10] is a special case with $c_R = p = 1$ and $c_T = \gamma$. Suppose $c_T \geq \gamma$; we observe that for a degenerate metric $d_\pi = 0$, the corresponding value function is a constant $V^\pi(\mathbf{s}) = c$. We will discuss the dangers posed by this relation in Section 3.2.3.

**Theorem 2** (Generalized VFA bound). *Let rewards be bounded as $R \in [0, 1]$ and $\Phi : \mathcal{S} \to \widetilde{\mathcal{S}}$ be a function mapping states to a finite partitioning $\widetilde{\mathcal{S}}$ such that $\Phi(\mathbf{s}_i) = \Phi(\mathbf{s}_j) \Rightarrow d_\pi(\mathbf{s}_i, \mathbf{s}_j) \leq 2\epsilon$, which produces an aggregated MDP $\langle \widetilde{\mathcal{S}}, \mathcal{A}, \widetilde{\mathcal{P}}, \widetilde{R}, \widetilde{\rho}_0 \rangle$. For any $c_T \in [0, 1)$, let $\overline{\gamma} = \min(c_T, \gamma)$. Given A1,*

$$|V^\pi(\mathbf{s}) - \widetilde{V}^\pi(\Phi(\mathbf{s}))| \leq \frac{2\epsilon}{c_R(1 - \overline{\gamma})} + \frac{2(\gamma - \overline{\gamma})}{(1-\gamma)(1-c_T)}, \ \forall \mathbf{s} \in \mathcal{S}. \tag{8}$$

This result generalizes previous performance bounds for $V^*$ to non-optimal policies. Previous bounds also assumed $c_T \geq \gamma$, while we generalize to arbitrary $c_T \in [0, 1)$. Here, the second term of the upper bound characterizes the penalty paid for "myopic" bisimulation (i.e., $c_T < \gamma$) in VFA error guarantees. We further show in the next section that $c_T < \gamma$ may be desirable when approximate forward models are used (see Thm. 4). Indeed, in Appendix C, we not only connect large $c_T$ to high variance in embedding norms, but also find surprisingly strong results when empirically investigating the use of $c_T < \gamma$. We speculate that with sufficient modelling capacity and a critic $V(\phi)$ being trained based on $\gamma$, the metric space may hold information about the environment regarding both timescales $\gamma$ and $c_T$.

### 3.2.2 Bisimulation Metrics with Approximate Forward Dynamics

We next examine theoretical properties of the on-policy bisimulation distance, which turn out to constrain properties of the approximate transition model. This motivates an additional architectural constraint, resulting in empirical improvements over prior work.

Metrics prior to [53] were defined based on ground-truth dynamics, which enabled the application of the Banach fixed-point theorem to show the existence of a unique metric. However, the Banach fixed-point theorem requires that a contraction mapping is applied on complete metric spaces. In this case, the metric space on which the Banach fixed-point theorem is applied is itself a space of metrics $\mathfrak{met}$ over states. For that reason, when bisimulation metrics were generalized from finite to continuous MDPs, a compact[4] state space was assumed to ensure completeness of $\mathfrak{met}$ [14]. In practice, when using an approximate forward dynamics model $\widehat{\mathcal{P}} : \mathcal{S} \times \mathcal{A} \to \mathcal{M}(\mathcal{S}')$[5], if compactness

---

[4]A metric space is compact if and only if it is totally bounded and complete [14].

[5]$\mathcal{M}(\mathcal{X})$ denotes the space of all probability distributions over $\mathcal{X}$.

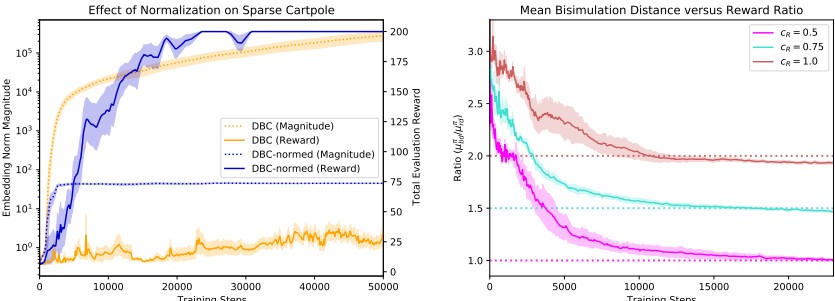

Figure 1: Theoretical properties under the SparseCartpole task. (**Left**) Embedding explosion occurs for DBC [53] for the sparse 2D-Cartpole task, while our constraints discussed in Sec. 3.2.2 keep the model stable. Our approach achieves maximum returns at evaluation, while DBC diverges. (**Right**) Given $c_T = 0.5$, Eq. (17) correctly predicts the ratio $\mu_{bd}^\pi / \mu_{rd}^\pi$ between bisimulation distance and difference of rewards at convergence. Dashed lines indicate analytically calculated targets, while solid lines correspond to mini-batch estimates of the ratio during training.

is not guaranteed, the same convergence guarantees do not apply. Thus, ideally, $\mathcal{S}'$ is a compact subset of $\mathcal{S}$. Here, we formalize these restrictions on latent representations and the forward dynamics model to guarantee convergence to a unique metric. The following states a constraint on the BSM solution space given approximate dynamics.

**Lemma 1** (Diameter of $\mathcal{S}$ is bounded). *Let* $d : \mathcal{S} \times \mathcal{S} \to [0, \infty)$ *be any bisimulation metric:*

$$\mathrm{diam}(\mathcal{S}; d) \coloneqq \sup_{\mathbf{s}_i, \mathbf{s}_j \in \mathcal{S} \times \mathcal{S}} d(\mathbf{s}_i, \mathbf{s}_j) \leq \frac{c_R}{1 - c_T}(R_{\max} - R_{\min}). \tag{9}$$

The above lemma holds for all bisimulation metrics (on-policy or otherwise) that are based on exact dynamics, if said metrics exist. Let us also consider an on-policy BSM with imperfect dynamics, such that it may not necessarily satisfy Lemma 1.

**Definition 3** (On-policy bisimulation metric with approximate dynamics). *Given an approximate dynamics model* $\widehat{\mathcal{P}} : \mathcal{S} \times \mathcal{A} \to \mathcal{M}(\mathcal{S}')$, $c_T \in [0, 1)$, *and* $c_R \in [0, \infty)$:

$$\widehat{d}_\pi(\mathbf{s}_i, \mathbf{s}_j) \coloneqq c_R|r_i^\pi - r_j^\pi| + c_T W_1(\widehat{d}_\pi)(\widehat{\mathcal{P}}^\pi(\cdot|\mathbf{s}_i), \widehat{\mathcal{P}}^\pi(\cdot|\mathbf{s}_j)). \tag{10}$$

Next, we provide a sufficient condition for the existence of a unique metric $\widehat{d}_\pi$ based on approximate dynamics, which also satisfies the upper-bound of Lemma 1. Meeting this condition shrinks the solution space of metrics being searched to a set known to contain $d_\pi$.

**Theorem 3** (Boundedness condition for convergence). *Assume* $\mathcal{S}$ *is compact. If the support of an approximate dynamics model* $\widehat{\mathcal{P}}$, *i.e.,* $\mathcal{S}' = \mathrm{supp}(\widehat{\mathcal{P}})$, *is a closed subset of* $\mathcal{S}$, *then there exists a unique on-policy bisimulation metric* $\widehat{d}_\pi$ *of the form Eq. (10), and this metric is bounded:*

$$\mathrm{supp}(\widehat{\mathcal{P}}) \subseteq \mathcal{S} \Rightarrow \mathrm{diam}(\mathcal{S}; \widehat{d}_\pi) \leq \frac{c_R}{1 - c_T}(R_{\max} - R_{\min}). \tag{11}$$

If this condition is not satisfied at any point during training, the system may diverge. Indeed, we confirm empirically in Sec. 4 that the absence of a norm constraint on the forward dynamics model may result in embedding explosion with practical consequences (see Fig. 1). Such explosions are due to compactness violations of the approximate dynamics (e.g., if the predictions always increase the embedding space diameter, the recursive nature of the metric can lead to runaway expansion). Luckily, the condition can be satisfied with ease, e.g., by projecting larger vectors onto the surface of a closed ball, $\mathbb{B}_c$, with diameter given in Lemma 1, such that the following are true:

$$\phi(\mathbf{s}) \in \mathbb{B}_c = \{\mathbf{x} \in \mathbb{R}^n \mid \|\mathbf{x}\|_q \leq \frac{c_R(R_{\max} - R_{\min})}{2(1 - c_T)}\}, \ \forall \mathbf{s} \in \mathcal{S} \tag{12}$$

$$\widehat{\mathcal{P}}(\cdot|\phi(\mathbf{s}), \mathbf{a}) \in \mathcal{M}(\mathbb{B}_c), \ \forall (\mathbf{s}, \mathbf{a}) \in \mathcal{S} \times \mathcal{A}. \tag{13}$$

We find that the constraint applied here is mild yet effective, as evidenced by significantly improved performance and stability when the constraints are active (see Fig. 1).

Clearly, a necessary condition for $\widehat{d}_\pi \to d_\pi$ is an error-free dynamics model. A natural question that arises from this view concerns the degree to which modelling errors affect VFA bounds.

**Theorem 4** (VFA bound in terms of model error). *Consider the same conditions as in Theorem 2, except that $c_T \in [\gamma, 1)$, $p = 1$, and $\Phi(\mathbf{s}_i) = \Phi(\mathbf{s}_j) \Rightarrow \widehat{d}_{\pi,\phi}(\mathbf{s}_i, \mathbf{s}_j) = \|\phi(\mathbf{s}_i) - \phi(\mathbf{s}_j)\|_q \leq 2\,\widehat{\epsilon}.$ Then:*

$$|V^\pi(\mathbf{s}) - \widetilde{V}^\pi(\Phi(\mathbf{s}))| \leq \frac{1}{c_R(1-\gamma)} \left( 2\,\widehat{\epsilon} + \mathcal{E}_\phi + \frac{2c_R}{1-c_T}\mathcal{E}_r + \frac{2c_T}{1-c_T}\mathcal{E}_\mathcal{P} \right), \forall \mathbf{s} \in \mathcal{S}. \quad (14)$$

*where $\mathcal{E}_\phi := \|\widehat{d}_{\pi,\phi} - \widehat{d}_\pi\|_\infty$ is the metric learning error, $\mathcal{E}_r := \|\widehat{r}^\pi - r^\pi\|_\infty$ is the reward approximation error, and $\mathcal{E}_\mathcal{P} := \sup_{\mathbf{s} \in \mathcal{S}} W_1(d_\pi)(\mathcal{P}^\pi(\cdot|\mathbf{s}), \widehat{\mathcal{P}}^\pi(\cdot|\mathbf{s}))$ is the state transition model error.*

See Appendix D for details, including a generalized version for $c_T \in [0, 1)$, listed as Corollary 3. Consider an ideal case where ground-truth rewards $r^\pi$ are available and the metric is learned perfectly (i.e., $\mathcal{E}_\phi = \mathcal{E}_r = 0$). Then, we observe that the choice of $c_T$ defines a trade-off between VFA error due to (a) forward model error $\mathcal{E}_\mathcal{P}$, and (b) "myopic" bisimulation (see Thm. 2, $c_T < \gamma$), where the BSM timescale is shorter than that of the discounted return.

### 3.2.3 On the Dangers of On-policy Bisimulation

Suppose that at training time, $|r_i^\pi - r_j^\pi| = 0$ for all pairs of states, e.g., during early training in a sparse reward setting. Then, unlike the policy-independent bisimulation metric, the on-policy formulation has a degenerate solution at $\mathrm{diam}(\mathcal{S}; d_\pi) = 0$, regardless of the structure of the underlying MDP.

**Lemma 2** (A reason for caution in on-policy bisimulation). *On-policy bisimulation metrics of the form Eq. (6) have an upper bound determined by their policy:*

$$\mathrm{diam}(\mathcal{S}; d_\pi) \leq \frac{c_R}{1 - c_T} \sup_{i,j} |r_i^\pi - r_j^\pi|. \quad (15)$$

Due to policy-dependence, the target metric $d_\pi$ changes with each policy update during joint training. As a result of this difficulty, convergence to a unique fixed point (i.e., a unique metric) for a learning algorithm was previously guaranteed with a strong assumption: "a policy that is continuously improving", as in Thm. 1 of [53]. Informally, if the policy is assumed to be continuously improving, in the worst case, the metric will have a fixed point $d_{\pi^*}$ after the policy itself reaches a fixed point, namely, the optimal policy $\pi^*$. However, this assumption may be too strong to guarantee convergence especially for continuous MDPs, as the policy learning process depends heavily on the encoder.

Specifically, to prove their Thm. 1, Zhang et al. [53] relied on the policy improvement theorem. For continuous MDPs, in practice, a policy $\pi(\mathbf{a}|\phi(\mathbf{s}))$ is learned via non-convex optimization (e.g., policy gradients, VFA), rather than the vanilla policy improvement algorithm. Thus, if the bisimulation metric is degenerate, a continuously improving policy cannot be guaranteed. As we will show, on-policy bisimulation metrics can in fact obstruct policy search in some cases (e.g., sparse rewards, low dispersion[6] rewards) by yielding a collapsed or exploded metric space, which is unable to approximate the value function. Specifically, we can define and relate measures of (i) collapse in the embedding space, and (ii) statistical dispersion of rewards under the current policy.

**Definition 4** (Measuring collapse and sparse rewards). *Let $\rho^\pi$ denote the stationary distribution over states, and $\nu^\pi$ the distribution over pairs of states, $(\mathbf{s}_i, \mathbf{s}_j)$ sampled independently from $\rho^\pi$. Then;*

$$\mu_{bd}^\pi := \mathbb{E}_{(\mathbf{s}_i, \mathbf{s}_j) \sim \nu^\pi}[d_\pi(\mathbf{s}_i, \mathbf{s}_j)] \qquad \mu_{rd}^\pi := \mathbb{E}_{(\mathbf{s}_i, \mathbf{s}_j) \sim \nu^\pi}[|r_i^\pi - r_j^\pi|]. \quad (16)$$

In the low dispersion case, $\mu_{rd}^\pi \approx 0$, meaning the current reward signal under $\pi$ is uninformative. However, this can have dire consequences for our embedding, as shown in the lemma below.

**Lemma 3** (Relating collapse and low-dispersion rewards). *Assume deterministic transitions and the existence of a stationary distribution $\rho_\pi$ over states. Given a bisimulation metric of the form Eq. (6):*

$$\mu_{bd}^\pi = \frac{c_R}{1 - c_T} \mu_{rd}^\pi. \quad (17)$$

---

[6]Statistical dispersion is an umbrella term used to describe measures of variability or diversity (e.g., variance).

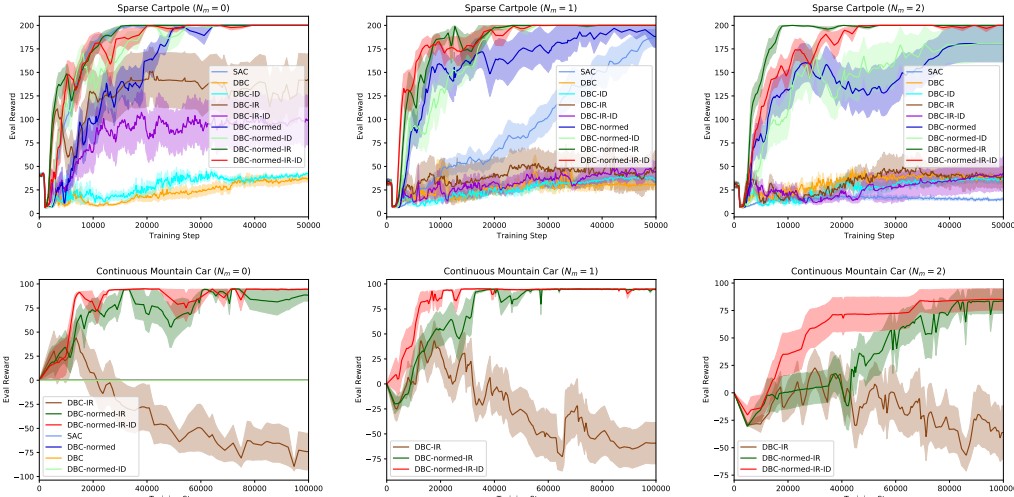

Figure 2: Results on the *modified* Gym tasks: Cartpole (**top row**) and Mountain Car (**bottom row**). The left, middle, and right columns have $0$, $\dim(\mathcal{S})$, and $2\dim(\mathcal{S})$ distractor noise dimensions respectively. We show the Soft Actor-Critic (*SAC*) and Deep Bisimulation for Control (*DBC*) baselines, along with our modifications using embedding normalization (*normed*), intrinsic rewards (*IR*), and inverse dynamics (*ID*). DBC struggles under all conditions of sparsity; while SAC handles it better (top-left), it cannot deal with high distraction (top-right) or more extreme sparsity (bottom row). However, latent normalization immediately improves performance (top row). Further, combining it with IR or IR+ID improves performance on all tasks. Shading shows standard error over 10 seeds.

Clearly, a collapsed state encoder, $\phi^*(\mathbf{s}_i) = \phi_0$, loses all information about the state. These observations motivate us to extend the method with inverse dynamics-based regularization and intrinsic rewards based on forward prediction errors (see Sec 3.3), since they promote $\mu_{bd}^{\pi}, \mu_{rd}^{\pi} > 0$ respectively. For empirical evidence of the relation in Lemma 3 at training-time, see Figure 1. Relationships between variances are also discussed in Appendix C.

### 3.3 Intrinsic Rewards and Inverse Dynamics

The core principle of DBC is to construct a representation such that a given $\phi(\mathbf{s})$ relates to other latent states via the PBSM, thus providing robustness to distractors, but also ensuring the latent state holds sufficient information to maximize return. As already discussed, however, uninformative rewards can induce metric learning issues for DBC, causing the embedding to hold insufficient information to solve the task. We consider two approaches to improving upon this representational deficiency: (1) using intrinsic rewards (IR), and (2) regularizing the latent space with inverse dynamics (ID) learning.

**Curiosity-Driven Intrinsic Rewards** One technique for encouraging exploration in sparse-reward environments uses intrinsic motivation via self-generated rewards, based on a notion of curiosity [8, 4]. In such cases, the agent rewards itself for entering unpredictable areas of the environment: in particular, at every time step $t$, we redefine the reward signal to be $r_t = r_{I,t} + r_{E,t}$, where $r_{E,t}$ is the extrinsic reward (i.e., the original environmental reward) and $r_{I,t}$ is the IR. Following prior work [8, 36, 43], we use the forward model error in the latent space to compute intrinsic rewards: $\widetilde{r}_{I,t} := \eta_r ||\widehat{\phi}_\mu(\mathbf{s}_t, \mathbf{a}_t) - \phi(\mathbf{s}_{t+1})||_2^2/(2n)$, where $\widehat{\phi}_\mu(\mathbf{s}_t, \mathbf{a}_t) = \mathbb{E}_{\phi(\mathbf{s}_{t+1}) \sim \widehat{\mathcal{P}}(\cdot | \phi(\mathbf{s}_t), \mathbf{a}_t)}[\phi(\mathbf{s}_{t+1})]$ is the mean of the predicted distribution from the forward dynamics model and $\eta_r > 0$ is a hyper-parameter controlling the intrinsic reward scale. We finally clamp to a fixed maximum $R_{\max,I}$, so that $r_{I,t} := \min(R_{\max,I}, \widetilde{r}_{I,t})$. We already train $\widehat{\mathcal{P}}$ for the DBC loss, so the additional computational cost is limited. Finally, note that IRs cause the reward signal to become non-stationary, resulting in the target of the BSM learning changing as well; however, as the agent explores and can better predict its environment over time, the IR (and thus its influence on the metric learning process) should fade.

**Latent Inverse Dynamics Prior** We also consider regularizing the learned latent state space, by having it learn to predict the inverse dynamics of the world, following prior work [3, 36]. Let $g_I : \mathbb{R}^n \times \mathbb{R}^n \to \mathcal{A}$ be an inverse model, which predicts the action $\mathbf{a}_t \in \mathcal{A}$ that changed $\mathbf{s}_t$ to

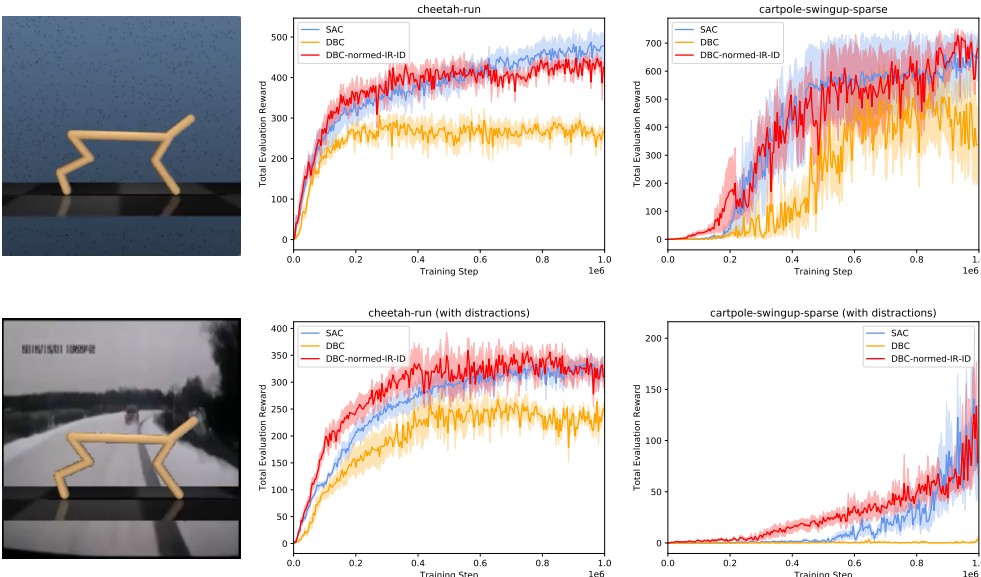

Figure 3: Performance on 3D robotics tasks from DMC [45] with 5 random seeds. **Leftmost column** illustrates the training setup for the `cheetah-run` task without (top) and with (bottom) natural video distractions. (**Top-left**) Our method significantly improves upon DBC [53] and is slightly below SAC [23]. (**Top-right**) DBC struggles under sparse rewards, while our approach remains robust to sparsity. (**Bottom-left**) Our approach significantly outperforms DBC and is more sample-efficient than SAC. (**Bottom-right**) DBC makes no progress under both sparsity and distractions, and SAC begins learning later due to sparsity.

$\mathbf{s}_{t+1}$ using the learned latent space: $\widehat{\mathbf{a}}_t \coloneqq g_I(\phi(\mathbf{s}_t), \phi(\mathbf{s}_{t+1}))$. This can be trained from observed transitions $(\mathbf{s}_t, \mathbf{a}_t, \mathbf{s}_{t+1})$, via $J_d(\phi, g_I; \eta_d) \coloneqq \eta_d ||\mathbf{a}_t - \widehat{\mathbf{a}}_t||_1 / n_a$, where $\eta_d > 0$ weighs the ID loss importance, and $\mathcal{A} = [-1, 1]^{n_a}$ for continuous control tasks. This loss prevents removal of information that pertains to the agent's ability to control the environment, but it is balanced against the loss driving $\phi$ to mimic the BSM. In this sense, we are placing a regularizing prior on $\phi$ via an auxiliary task, such that it adheres to the BSM requirement when possible, but still transmits useful information in the sparse reward case instead of collapsing. Yet, since only *actions* need be predicted, distractor aspects of the observation that the agent cannot influence are naturally ignored. Hence, we expect less disruption to the BSM encoding than other auxiliary tasks (e.g., reconstruction).

**Summary** In this section, we (1) provided generalized bounds connecting the value function to the on-policy BSM (Sec. 3.2.1); (2) showed that the BSM satisfies a fixed latent diameter, and devised a normalization for enforcing this property during learning (Sec. 3.2.2), which not only prevents embedding explosion but improves performance (Fig. 1); and (3) found the PBSM is susceptible to embedding collapse with sparse rewards, and suggested IR and ID to mitigate it (Sec. 3.2.3, 3.3).

## 4 Experiments

In this section, we seek answers to the following questions concerning our main hypotheses:

1. Do the embedding collapse and explosion issues predicted theoretically occur in practice?
2. Do our contributions address these problems?
3. Does our proposed approach preserve the noise-invariance property of bisimulation?
4. How do our proposed improvements interact with each other?
5. How does our method perform compared to prior work, particularly with sparse rewards?

To that end, we experiment on several altered classic control tasks from OpenAI Gym [7] by (i) sparsifying the reward signal, and (ii) augmenting the environment state with noisy dimensions, to simulate distractions. We also perform larger scale experiments on two challenging vision-based 3D robotics benchmarks from the DeepMind Control Suite [45].

**Sparse Noisy Cartpole**  First, we modified the `Cartpole-v0` task, in which an agent tries to keep a pole upright. To increase sparsity, we shrank the angular extent in which the pole must be to earn rewards to 1% of its standard value. Then, to mimic distractors, we concatenate an $N_m \dim(\mathcal{S})$-dimensional vector sampled from an isotropic Gaussian to the state vector. (See Appendix E.1.1 for more details.) The performance of the respective models, as well as other baselines are shown in Fig. 2, top row. While DBC struggles on this task due to the sparsity, the Soft Actor-Critic (SAC, [23]) baseline, which does not use BSMs, is able to solve the problem (top-left inset). Our proposed modifications address the embedding explosion problem and perform on-par with the SAC baseline, both when combined and separately. Furthermore, when distractors are introduced (top-middle and top-right insets), SAC also fails, while our approach combining intrinsic rewards (IR) and robust metric learning still solves the task. See also Appendix E.1.4 for results with $N_m = 3$.

**Noisy Mountain Car**  We also consider the classic `MountainCarContinuous-v0` task [32], in its continuous control form (see Appendix E.1.2 for details). Since the task was already highly sparse, we modified only the distraction aspect, as in the Cartpole scenario. As shown in Fig. 2, all methods without IR are unable to solve the task and are stuck with rewards close to zero. However, DBC with IR is unstable and unable to solve the task either. Only using the normalization allows the agent to succeed; furthermore, the inverse dynamics (ID) regularization improves convergence speed and stability. This trend continues at even higher distraction levels as well (see Appendix E.1.4).

**Sparse Noisy Pendulum**  We also modified the continuous `Pendulum-v0` task to have a higher degree of sparsity and distraction. Due to space constraints, we relegate details to Appendix E.1.3. We find that our method performs on par with DBC, but outperforms SAC in the presence of distractions.

**DeepMind Control Suite**  The DeepMind Control Suite (DMC) [45] contains a set of 3D robotics tasks based on the MuJoCo physics simulator [47]. We include results from two DMC tasks, namely, `cheetah-run` ($\dim(\mathcal{S})=18, \dim(\mathcal{A})=6$) and `cartpole-swingup-sparse` ($\dim(\mathcal{S})=4, \dim(\mathcal{A})=1$), shown in Figure 3. The former is a common benchmark where Zhang et al. [53] showed that their method performed sub-par in the *absence* of distractors. The latter is a task we select due to reward sparsity. Similarly to [53], we train for each task with and without natural video distractions, as illustrated in Figure 3. In all cases, our approach performs significantly better than the DBC baseline [53]. Without distractors, our method performs competitively with SAC [23], but when distractions are introduced, our approach is slightly more sample-efficient. With these experiments, we verify that our improvements carry over to larger-scale tasks where learning is over raw-pixels.

## 5 Conclusion

**Limitations and Impact**  One shortcoming of our approach is the lack of a principled way to set hyper-parameters for IR and ID, which was done empirically. Indeed, our use of forward model error and ID regularization itself may not be optimal. Sometimes, we observe that the embeddings all attain norms close to the maximum allowed value, suggesting alternative approaches to normalization may be helpful. Finally, IR and ID incur an additional computational cost to training. In terms of broader impact, while our work is foundational RL, it focuses on removing distractions from internal representations. However, this culling can remove information that the agent deems unrelated to the task, potentially leading to unsafe decisions in unseen scenarios (e.g., for mission critical systems).

**Discussion**  In this work, we investigated the behavior of on-policy deep bisimulation metric learning approaches, which construct efficient neural representations that are invariant to noise and distractors, without reconstructing the raw input [10, 53]. We identified embedding collapse and explosion as potential failure modes of this method via theoretical analysis, and highlighted that it may be especially susceptible to failure in sparse reward settings. Our experiments confirmed the dangers of these failure modes and showed that our proposed remedies address the issue. In particular, we enforced a norm constraint on state representations, and incorporated intrinsic motivation and latent regularization in our technique. The resulting approach preserves the noise-invariance property of bisimulation metrics while comparing favorably against strong baselines [23, 53] on altered versions of classic control tasks, which we rendered more challenging by sparsifying the rewards and synthesizing distractors, as well as harder tasks with visual observations. Future work includes investigating alternative ways to improve embedding regularization and reward informativeness, as well as more realistic 3D robotics benchmarks, with sparse reward structures and heavy distraction.

**Acknowledgments** We thank Sven Dickinson, Allan Jepson, and Amir-massoud Farahmand for helpful discussions. The support of NSERC (CGSD3-534955-2019) and Samsung AI Research is gratefully acknowledged.

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
