# OpenReview forum: "Towards Robust Bisimulation Metric Learning"
_NeurIPS.cc/2021/Conference — NeurIPS 2021 Poster_

### Official Review · Reviewer_mSYo · 2021-07-16

**Rating:** 7
**Confidence:** 3

**Summary:**

This paper considers the problem of learning good representations for reinforcement learning problems. It approaches this from the perspective of bisimulation metrics, extending earlier work on on-policy bisimulation for control (DBC). The paper identifies issues with DBC that also appear in other types of metric learning, and includes a theoretical analysis of why this happens. To remedy the issues the paper proposes normalizing the representations and including an extension to DBC by including an inverse dynamics model and intrinsic rewards.

**Limitations And Societal Impact:**

#### Limitations:
Yes, the authors discuss limitations of their work.
#### Societal:
Yes, the authors discuss potential societal impact. A minor suggestion would be to add a comment on how improving RL (as this paper aims to do!) could impact society, e.g. in terms of labor markets or autonomous weapons.

**Main Review:**

### Strengths:

* This paper is well written and answers many of the questions that arise while reading it immediately.
* The analysis of certain issues in DBC is very nice to have. The theoretical section was (mostly) easy to follow and was informative.
* The theory section has a clear added value in analysing the issues with DBC and proposing improvements. For example, compact spaces are assumed for continuous MDPs and this assumption is violated in DBC. This motivates the authors to propose normalizing the representations to a closed ball. Additionally, there is a degenerate solution when rewards are sparse, which motivates the authors to propose intrinsic rewards to prevent this from occurring.
* An ablation study is included (though it is incomplete)
* The paper does a good job of reviewing the related work, only missing a few related references (included below).

### Weaknesses:
* The choice of inverse dynamics as a regularizer is not well-motivated. This choice is not grounded in the same way that the rest of the paper is. For example, why choose an inverse dynamics model and not a different self-supervised loss? Why not a different combination of approaches? The motivation given for inverse dynamics could also be applied to reconstructions or contrastive coding. Additionally, there are no experimental results showing that an inverse dynamics model is best; as such this choice is the weakest part of the paper.
* As there is no experimental comparison to other additions to DBC (e.g. different self-supervised learning approaches), it is unclear why  inverse dynamics is the preferred method to solving the issues with DBC.
* The paper focuses heavily on one method; DBC, and as such is somewhat limited in scope.
* The paper, while well-written, is not fully self-contained - certain terms are used without being defined. A few examples:
  * What is a ‘compact subset of $\mathcal{S}$’?
  * What is a diameter of a state space?
  * What is sup? The supremum?
  * What is supp? Also the supremum?
  * What is $\mathcal{H}$ in line 259?
  * What is a low dispersion reward?
* According to the paper, DBC suffers from common metric learning problems, but there are no references included to other metric learning papers that exhibit similar issues.
* If the intrinsic reward is based on the forward model error, it should not be useful under state collapse, as the error would be 0.
* The non stationarity introduced by IR might be an issue in more complex environments.
* Line 294: Claim: “Hence, we expect less disruption to the BSM encoding than other auxiliary tasks (e.g. reconstruction)”. Reconstruction is not the only possible auxiliary task. What happens when using a better auxiliary loss, such as action-dependent contrastive coding?
* Ablation study is incomplete -- DBC-ID and DBC-normed-ID are not included.
* “(Left) Our method significantly improves upon DBC [36] and is on par with SAC” this looks like it isn’t true, as SAC achieves a higher performance in the end.
* The method isn’t evaluated on the same benchmarks as DBC, with comparable levels of distraction (e.g. natural video). As a result, it is unclear what the performance improvement would be in those settings.


### Questions:
* “two states that only differ in rewards far in the future should still not be bisimilar” - Standard bisimilarity is defined over immediate, not future rewards. Could you elaborate?
* The assumption that the reward is normalized between 0 and 1 -- is this taken into account in the experiments?
* The sentences 193-195 are unclear, could you clarify?
* Eq. 11: x \in R^d is this d overloaded?
* Eq. 11: what is B_c? The ball with radius c? It looks like c is not defined.
* Eq. 12: Does M(B_c) mean the space of probability distributions over the ball with radius c?
* Lemma 3 assumes a stationary distribution over states, but this distribution changes as the policy changes. How does this influence the theory/results?
* Figure 1: what is magnitude? The magnitude of the embedding?

### Minor comments:
* Below are some suggested references on state representation learning that could be relevant to include in related work.
* Plots are hard to read in black and white
* While the motivation for requiring more informative rewards is clear (as mentioned in strengths), it is not motivated why this takes the form of intrinsic rewards, as compared to e.g. reward propagation.
* SAC does not fail in sparse-cartpole with N_m=1, it just takes longer to converge


### Overall:

The score is mostly based on the following:
* The choice of inverse dynamics (vs action-dependent contrastive coding or another type of auxiliary loss) should be properly motivated, either theoretically or experimentally.
* Missing references should be included
* Evaluation should be performed on the same benchmarks as DBC, e.g. with natural video distraction.

I would raise the score if these are addressed.

#### Missing references:
* Biza et al, "Online Abstraction with MDP Homomorphisms for Deep Learning"
* Grimm et al, "The value equivalence principle for model-based reinforcement learning"
* Le Lan et al, "Metrics and continuity in reinforcement learning"
* Ravindran & Barto, "Approximate Homomorphisms: A framework for non-exact minimization in Markov Decision Processes"
* Taylor et al, "Bounding Performance Loss in Approximate MDP Homomorphisms"
* Van der Pol et al, "Plannable Approximations to MDP homomorphisms: equivariance under actions"

#### Suggested references
* Agarwal et al, "Contrastive Behavioral Similarity Embeddings for Generalization in Reinforcement Learning"
* Corneil et al,  “Efficient model-based deep reinforcement learning with variational state tabulation”
* Dadashi et al, Offline Reinforcement Learning with Pseudometric Learning
* Francois-Lavet et al, “Combined reinforcement learning via abstract representations”
* Watter et al,  “Embed to control: A locally linear latent dynamics model for control from raw images”

#### Update
Thank you for the rebuttal, and for the update on the experiments. I think adding the new experiments to the paper will make the paper more convincing.
When reviewing, I only had a few issues with this paper: the lack of motivation for the choice of inverse dynamics (as opposed to another auxiliary loss), the missing evaluations on benchmarks with natural video distractions (for a fair comparison to DBC), the somewhat limited scope (due to the focus on DBC), some missing references, and the paper not being fully self-contained.
In the author response, the authors motivate their choice of inverse dynamics, clarify that their results can be useful for bisimulation metric learning in general, promise to add the references and amend the paper where it is unclear, and in a second reply give an update on benchmarks with natural video distractions, with positive results.

The authors thus address my original concerns. However, reviewer pEeB raised concerns about the theoretical analysis that are important to address. I have therefore decided keep my original score.


**Time Spent Reviewing:**

4

---

> ### Author Response · Authors · 2021-08-10
> **Official Comment for Reviewer mSYo**
>
> We thank the reviewer for their time and detailed feedback. The reviewer highlighted that they found our paper to be well-written with an easy-to-follow and informative theoretical section. We are also glad to see they described our analysis of issues with DBC as having clear added value.
>
> Weaknesses:
>
> - Motivation for the use of inverse dynamics and its evaluation: We agree with the reviewer that the representation learning literature has a rich variety of powerful methods, which could potentially replace the inverse dynamics (ID) regularizer. Indeed, as noted in the Limitations section (L340), the use of intrinsic rewards (IR) and ID are not necessarily theoretically optimal, in the bisimulation metric context. However, we nevertheless felt that ID was a natural choice for our choice of setting (sparse rewards and high distraction). First, ID is naturally robust to observational distractors, as predicting the action in the presence of distraction requires learning to ignore irrelevant information (e.g., [2]). Second, given an encoder, the ID method is computationally efficient, with only light overhead (as $g_I$ is relatively small). Third, since it does not rely on the reward signal, it may still provide a useful representation even in the sparse-reward scenario, preventing embedding pathology. Indeed, Pathak et al. [27] utilize ID in conjunction with IR, in sparse-reward environments with additional distractors, suggesting it is a proven approach under such conditions. Regarding auxiliary losses, while contrastive losses avoid reconstruction, Zhang et al [36] find that such approaches are less robust than DBC. More sophisticated techniques, such as action-dependent CPC, may very well overcome such issues. However, we chose to focus on ID, for the reasons above, and leave further investigation of the optimal embedding regularizer to future work, noting our approach did improve significantly over DBC.
>
> - Limited scope due to focus on DBC: we agree that our model architecture and training algorithm build heavily on DBC and that the potential weaknesses of DBC have inspired a significant portion of this work. However, we would like to remark that our main theoretical results in Theorems 1-4 are broadly applicable to bisimulation metric learning whether or not the DBC algorithm in particular is used for training. We hope that these will inspire future work with new algorithms and architectures by extending the community’s understanding of bisimulation in general and on-policy bisimulation.
>
> - We will add the following information to the manuscript to improve readability:
>
>    - A compact subset of S is one that is totally bounded and complete.
>    - The diameter of a state space is the maximum distance between any two states measured via a distance function (see Lemma 1).
>    - $\sup$ is the supremum.
>    - $\mathrm{supp}(p)$ is the support of the distribution of $p$.
>    - $\mathcal{H}$ denotes entropy, i.e., the expected Shannon information content.
>    - Statistical dispersion is an umbrella term used to describe measures of variability (e.g., variance).
>
> - We will add the following metric learning papers for reference to discussion of collapse and explosion issues:
>
>    - Hermans, Alexander, Lucas Beyer, and Bastian Leibe. "In defense of the triplet loss for person re-identification."
>
>    - Schroff, Florian, Dmitry Kalenichenko, and James Philbin. "Facenet: A unified embedding for face recognition and clustering."
>
>    - Wu, Chao-Yuan, R. Manmatha, Alexander J. Smola, and Philipp Krahenbuhl. "Sampling matters in deep embedding learning."
>
> - While the reviewer is correct that a completely collapsed embedding will not provide useful intrinsic rewards, the goal is to utilize such rewards to prevent such a collapse in the first place, by allowing the bisimulation metric to differentiate between states, based on the agent's surprise. In addition, the use of inverse dynamics (ID) regularization, is designed to prevent collapse directly (by requiring the embedded vectors to solve an independent task); this may also have the effect of ensuring the IR remains functional as well.
>
> - We agree that the non-stationarity of the reward could cause more of an issue in more complex environments. Our assumption is that, even for more complicated tasks, the agent will eventually begin to better predict the world. This is made more plausible by the fact that the agent need not be able to predict full observations, but rather need only predict embeddings that are specifically designed to care only about task (and/or action) related information. We additionally note that the IR clamped maximum $R_{\mathrm{max},I}$, as well as the IR weight scale coefficient $\eta_r$, are both used to preserve stability by ensuring the IR does not overwhelm the agent.
>
> - Regarding the completeness of the ablation study, the reviewer correctly notes that “DBC-ID and DBC-normed-ID are not included” in the ablation analysis. We found DBC-ID and DBC-normed-ID to perform fairly similarly to DBC and DBC-normed, respectively. We will include these additional settings in the plots for the final version.
>
> - Comment on Fig.3: we will replace “on par” with “slightly below”.
>
> - Natural video distractions: following the reviewer’s recommendation, we are currently performing experiments for DMC with natural video distractions in the background. We have obtained preliminary results on the cheetah-run task, which show better results compared to both SAC and DBC. We intend to add these experiments to the final manuscript.
>
> Questions:
>
> - Via the recursive relation induced by the bisimilarity transition distance term, differences in rewards in the future affect bisimilarity, although to a reduced extent. Here $c_T$ serves a role analogous to a discount factor.
>
> - Yes, $R \in [0, 1]$ will be added to the experiments section.
>
> - L193-195: Our remarks on L193-195 concern the common choice to assign the value of $\gamma$ to $c_T$. Both coefficients can be viewed as timescales: $\gamma$ defines the timescale of future rewards that the agent considers, while $c_T$ controls the effect of the recursive, forward-looking term in the bisimulation distance versus that of the immediate reward difference. Higher $c_T$ thus emphasizes the difference in the distribution of future states - shifting the “timescale” of the metric on states to be less myopic. While $\gamma$ is a property of the MDP, $c_T$ controls the learned representation; it is not obvious a priori that these two time-scales, therefore, should be identical. Indeed, decoupling them can be helpful to performance (L192). Lines 193-195 speculate that the state representation, given appropriate training, can hold information about rewards and dynamics relevant to both time scales, and that it is not necessary to set them equal to each other, as most prior methods do. We will endeavour to better clarify this in the paper.
>
> - Thanks, $d$ will be replaced with $n$ to avoid overloading.
>
> - $\mathbb{B}_c$ denotes “closed” ball (will be clarified). $\mathcal{M}(\mathbb{B}_c)$ is the space of distributions over it, given radius in Eq. 11.
>
> - Stationary distribution assumption in Lemma 3:
>
>    - The reviewer correctly notes that the stationary distribution in Lemma 3 is policy-dependent, and hence changes as training proceeds.
> Indeed, the lemma assumes a fixed policy, and thus computes its expectations over a fixed stationary distribution.
> If the agent sees states sampled from a different distribution, whether due to non-stationarity of the policy or some other reason, then the statistical relation in Lemma 3 may not be guaranteed to hold.
> Nevertheless, empirically, we find that for simple MDPs on which we tested, the relation approximately holds despite the continuously changing policy, especially later in training.
> Potentially, this may be due to the learning process (i.e., rate of policy change) slowing down later in training, as the agent and its representation improves. The use of a replay buffer and Polyak updates also reduce the non-stationarity of the policy.
>
> - Yes, magnitude is the magnitude of the embedding as in y-axis labels.
>
> Minor comments:
>
> - Thanks for pointing out, missing references will be added to related work.
>
> - We will try to alter the plots to improve visual accessibility with different dash types.
>
> - Motivation for intrinsic rewards (IR): as in the case of inverse dynamics mentioned previously, we do not wish to claim that curiosity-driven intrinsic rewards, based on forward model error, are the optimal approach to preventing embedding pathologies in the latent bisimulation metric space (e.g., as mentioned in Limitations, L340-341). However, given that sparse rewards are a cause of such problems, it seems natural that adding an informative intrinsic reward could be a potential solution. Furthermore, prior work has shown that such curiosity-based approaches to exploration have proven useful in sparse environments, including the case of high distraction [27], in which the bisimulation approach is designed to operate. In addition, since computing bisimulation distances already requires a forward model, there is minimal effect on computational efficiency from adding IR (L280-281). Nevertheless, other approaches of greater complexity, such as reward propagation via potential functions, may prove to be more effective. We will add further commentary on the potential of such alternative methods in the paper.
>
> - “top-middle and top-right” will be replaced with “top-right”. We will reword to say SAC fails under “heavy” distraction.

---

> > ### Author Response · Authors · 2021-08-23
> > **Update on Additional Experiments**
> >
> > We have run our method on the larger scale DMC tasks in the paper, but now altered to include natural video distractions as in the DBC paper. Our results show that we significantly outperform DBC, which fails to obtain any rewards in cartpole-swingup-sparse and remains under 250 reward for cheetah-run, while we exceed 300. Compared to SAC, while our final performance is similar, we have superior sample efficiency: our approach reaches peak performance before 400K steps, while SAC reaches it much later (after 800K) for cheetah-run, while for cartpole-swingup-sparse, we begin rising above zero around 250K,  while SAC starts around 450K, only catching up after 800K.

---

### Official Review · Reviewer_FWDg · 2021-07-21

**Rating:** 7
**Confidence:** 2

**Summary:**

In this paper, the author proposed a robust bisimulation metric that could combat the collapse and explosion issues. The authors analyzed the cause of these issues theoretically and further proposed  a set of remedies including a norm constraint on the representation space, and intrinsic rewards and latent space regularization. The authors did experiments in the gym environment to verify the proposed methods.

**Main Review:**

Pros:
1. The authors modify the previous metrics and provide a generalized metric. The analysis of the new metric compared with the old one is solid.
2. In the experiments, the newly proposed method shows better robustness against sparsity compared with DBC.
3. The authors further show the policy-dependence may cause caution collapse in on-policy bisimulation.

Cons:
1. The experiments have simple/similar dynamics. It would be of beneficial for readers to know whether the approach would still gain obvious edges when the dynamics of the task is complex, such as a sparse version of the spider or walker in gym. Now the main challenge seems only coming from the dimension of the noise.

**Time Spent Reviewing:**

10

---

> ### Author Response · Authors · 2021-08-10
> **Official Comment for Reviewer FWDg**
>
> We thank the reviewer for their time and feedback. To ensure that the improvements introduced by our method carry over to more complex environments and dynamics, we performed experiments on the cheetah-run and cartpole-swingup-sparse tasks of the DeepMind Control Suite (see Fig. 3). These are 3D environments simulated in MuJoCo and the RL agent learns from raw pixel observations in these cases. In particular, the cheetah-run environment has complex dynamics, with an 18D state space for the simulated robot’s joint positions and velocities, as well as a 6D action space (this information will be added to the final manuscript). We found that our improvements over DBC indeed carry over to this more complex task.
>
> The DeepMind Control Suite offers a sparse-reward setting for a small subset of the available tasks including cartpole-swingup-sparse; unfortunately all of these tasks have simpler dynamics than tasks like cheetah-run and walker. While this task has simpler dynamics ($\mathrm{dim}(\mathcal{S})=4, \mathrm{dim}(\mathcal{A})=1$) compared to cheetah-run, the task is relatively more challenging than cartpole-balance-sparse, since the pole starts in downright position. However, we agree with the reviewer that further evaluation on even more complex tasks, possibly real robots, would strongly complement the theoretical contributions of our work, which we left for future efforts.

---

### Official Review · Reviewer_pEeB · 2021-07-24

**Rating:** 5
**Confidence:** 3

**Summary:**

The paper analyzes properties of bisimulation metrics, identifies potential issues in learning with sparse rewards, and proposes methods to fix the issues.

**Ethical Concerns:**

No.

**Limitations And Societal Impact:**

Parts of the limitations are mentioned in the paper, but most of the analysis are only on the existence of the bisimulation metric, and there is a big gap for practical algorithms to actually converge to the unique metric even if it exists.

**Main Review:**

Learning good latent representation is important for RL problems with high dimensional observations, and recent approaches based on bisimulation metrics have shown promising results. This paper analyzes the existence of unique on-policy bisimulation metrics and their connection to value function bounds. Based on the properties, the paper proposes a normalization method to fix some embedding issues of prior methods in sparse reward environments, and it further proposes improvements using intrinsic rewards and inverse dynamics.

The ideas to analyze on-policy bisumlation metrics is very reasonable, and the proposed methods seem to be both theoretical and practical appealing, but the paper (particular Section 3) is very difficult to read, and many key concepts are either missing or hard to follow. Below are some issues with the paper in the current status.

As the paper claimed, one of the key contributions is the convergence condition of bisimulation metric in learning. However, there is never a formal statement of what is the "sufficient condition for the existence of a unique bisimulation metric". This condition seems to be mentioned after Theorem 3, but the discussion is very confusing. From the statement of Theorem 3, "the sufficient conditions" seem to refer to cT<1, p=1, and the support of the approximate dynamics is closed. But later in (11) the main condition seems to refer to the boundedness of the metric. The boundedness of the unique metric is mentioned in Theorem 3, but as a property instead of a sufficient condition. After checking the proof of Theorem 3, the boundedness is indeed not used in proving the existence of the metric. So why does the boundedness of metric becomes the sufficient condition? Does the boundedness have any relationship to the closedness of the support? Even more confusing, in all discussions after Section 3 the condition is referred to "embedding normalization" without any definition nor explanation. Does normalization mean the requirements (11)? How does the normalization actually work in learning with parameterized embedding functions?

There are some missing definitions and typos which highly affect readability. There is no definition of the diameter before Lemma 1, no definition for the different forms of metrics used in Theorem 4 besides in the appendix. In the paragraph before Theorem 3, one metric d with hat but another one without hat. What is the metric d appearing in (11)? There is no indication on what the sup in (14) over on which doesn't seem to be over policies.

The idea of investigating the danger of on-policy bisimulation with sparse rewards in 3.2.3 is very interesting, but the discussion is very unclear and may need further explanation. The section claims that the challenges of degeneration is only an issue with on-policy bisimulation, but there is no comparison with policy-independent bisimulation. If policy-independent bisimulation doesn't have the issue, then why prior methods don't work? The paper also claims that this degeneration might be the reason for poor performance of DBC and the proposed methods fix the degeneration, but the simulation results actually show exploding behaviors instead of collapsing. How does the analysis explains exploding behaviors of DBC? The proposed intrinsic rewards and inverse dynamics methods seem reasonable, but there does not seem to have any connection to the analysis in 3.2.3 and why these methods can resolve the degeneration issue.

As said above, I like the ideas of the paper but would not recommend to accept it given the issues mentioned above. I will consider raising the score if the paper can be revised to address the above points.




**Time Spent Reviewing:**

15 hours

---

> ### Author Response · Authors · 2021-08-10
> **Official Comment for Reviewer pEeB**
>
> We are grateful to the reviewer for their detailed comments, and appreciate that they both "like the ideas" and feel the "proposed methods seem to be both theoretical and practical appealing".
>
> - Regarding sufficient conditions for the existence of a unique bisimulation metric, we will better clarify them and include the following information more clearly:
>
>   - The first mention of sufficient conditions for showing the existence of a unique bisimulation metric is in L132-134; one needs to ensure that the RHS is a contraction so that the Banach fixed point theorem can be applied. We also show how this proof unfolds in the proof of Remark 1 in Appendix B.
> Banach FP theorem requires that the space on which it is applied is “complete”. In this case, Banach FP theorem is applied on the space of metrics. To ensure that the space of metrics over states is complete, [11] showed that it is sufficient to assume that the state space itself is “compact”.
> However, [11] was assuming ground-truth dynamics, which implied that unrolling the transition dynamics would result in future states also being in that same compact state space.
> Using approximate dynamics, however, one needs to ensure that repeated application of transition dynamics $\widehat{\mathcal{P}}$ keeps the state in a compact space (i.e., ensuring bounded support), which leads us to Thm. 3.
>
> - > "So why does the boundedness of metric becomes the sufficient condition?"
>
>    - We differentiate between two conditions: (C1) the implication proven in Thm. 3, which shows that the support of $\widehat{\mathcal{P}}$ being contained in $\mathcal{S}$ is *sufficient* for metric uniqueness and existence and (C2) boundedness of the metric (a property of any BSM, as in Eq. 9, and enforced by us via Eq. 11, which is *necessary* to satisfy the (sufficient) condition (C1). Thus boundedness itself is not the sufficient condition referred to in L210, but it is necessary for it.
>
>    - Regarding the relation to our embedding normalization technique, Thm. 3 shows that (assuming the forward model outputs valid states in $\mathcal{S}$) the optimal metric derived from using that approximate forward model must be bounded according to Eq. 9 in Lemma 1. Hence, if we obtain a learned metric that is *not* bounded, then it is necessarily *not* the optimal metric for the given approximate dynamics. In other words, the contrapositive of the theorem tells us that boundedness (wrt Eq. 9) is necessary for correctness of the learned metric.
> This helps motivate our “embedding normalization” technique. Assuming that $\mathcal{S}$ is compact (and thus bounded), the true metric is bounded by Eq. 9. Hence, any learned or approximate metric that violates this bound cannot be the correct metric, and should be removed from the “solution space of metrics” (L211). Thus, we use our normalization technique to ensure that this bound is satisfied.
>
> - > "Does the boundedness have any relationship to the closedness of the support?"
>
>    - Yes, a closed subset of a bounded space is also bounded. Secondly, a closed subset of a compact space is also compact.
>
> - Regarding "embedding normalization", we would like to differentiate between (i) the desired property of the support of the forward model to be bounded when $\mathcal{S}$ is compact, compared to (ii) the technique we use to enforce the satisfaction of this property. Indeed, Eq. 11 is our technique for doing so. Thus, the normalization does indeed mean Eq. 11, but the actual condition we wish to satisfy is in Eq. 10, using our method given in Eq. 11. As to how this technique works with parameterized embedding functions, we simply use the Euclidean norm of the embedded vectors, including the normalization as part of the forward pass of the encoder.
>
> - Missing definitions and typos:
>
>    - Regarding the missing definitions, thank you, they will be added to the paper.
>    - "In the paragraph before Thm. 3, one metric $d$ with hat but another one without hat." The hat signifies the estimated (learned) metric, while the one without is the ground truth. We will add additional clarification text.
>    - The metric appearing in Eq. 11 is the learned Euclidean metric in the embedding phi space, which approximates the bisimulation metric. We will alter the text to clarify.
>    - The supremum in Eq. 14 is over pairs of per-state expected rewards (or, equivalently, over $i,j$). We will clarify the notation.
>
> - Regarding the danger of on-policy bisimulation:
>
>   - While policy-independent bisimulation does not necessarily have the same embedding issues, it introduces separate difficulties. In particular, it results in computational tractability challenges, as estimating the metric becomes too difficult for continuous action spaces due to the max operation. Further, as noted in our refs [7,8], policy-independent BSMs can be unnecessarily pessimistic (e.g., poor actions that no reasonable policy would choose can still greatly affect the state metric).
>
> -  > "How does the analysis explains exploding behaviors of DBC? "
>
>    - One theoretical reason for embedding explosions is the presence of a $1 / (1-c_T)$ term in the expected value of the distance, which can be seen in Eq. 16. Since we often set $c_T = \gamma \approx 1$, this term tends to amplify the distances and norms of the metric during learning. This is further discussed in more detail in Appendix C. We will move some of this explanation into the main paper.
>
>    - Another reason is the violation of the compactness condition by an approximate dynamics model as discussed in Sec 3.2.3. For example, consider an erroneous dynamics model that doubles the embedding space volume after each timestep. In such a case, due to the recursive definition of the bisimulation metric, pairwise distances between states will also grow after each fixed-point iteration, which may result in explosion.
>
> - > "The proposed intrinsic rewards and inverse dynamics methods seem reasonable, but there does not seem to have any connection to the analysis in 3.2.3 and why these methods can resolve the degeneration issue."
>
>    - We briefly motivate the connection of IR and ID to the analysis on L260-262: "These observations motivate us to extend the method with inverse dynamics-based regularization and intrinsic rewards based on forward prediction errors (see Sec 3.3), since they promote $\mu^\pi_{bd}, \mu^\pi _{rd} > 0$ respectively." Intuitively, the numerical embedding pathologies stem from uninformative rewards: adding IR should mitigate this root problem directly, though it assumes (as many curiosity-based works do) that exploration rewards can provide a useful representation. ID, on the other hand, provides a reward-independent regularization of the latent space, which has proven to provide a useful representation for distracted and sparse-reward environments in prior work [2,27]. Our motivation is that, even in the presence of uninformative rewards, the regularized embedding will still contain useful information. We will add further clarification of this to the paper.

---

### Official Review · Reviewer_XWno · 2021-07-28

**Rating:** 6
**Confidence:** 1

**Summary:**

This paper investigates learning better representations for deep reinforcement learning algorithms using bisimulation metric (BSM), such that the representations are more efficient and robust from distraction (e.g. noise). The proposed method improves the Deep Bisimulation for Control (DBC) algorithm for tasks with uninformative rewards and the authors show their method is competitive in those environments.



**Limitations And Societal Impact:**

See the main review.

**Main Review:**

This paper focuses on learning more robust and efficient representations for deep reinforcement learning algorithms. The analysis and resulting method build on previous work of the Deep Bisimulation for Control (DBC) algorithm but improves for environments with sparse rewards in particular. The authors analyzed theoretically embedding collapse and explosion could happen with sparse rewards and proposed remedies to fix them. The authors showed that the resulting algorithm was not only better with sparse rewards but also more stable when the observations are noisy. The topic is definitely closely related and significant to the community. Although it builds on DBC, the analysis and proposed algorithm are novel to the best of my knowledge. The paper is also well-written and easy to follow in general. Some questions: 1.can you talk more about embedding collapsing and exploration? 2.the experiments are conducted on sparse cartpole and continuous mountain cars, which are relatively low-dimension, maybe evaluate and compare the methods in MuJuCo tasks with sparse rewards? 3. fig.3, why the proposed method is a bit worse or almost the same as sac? could you explain more? especially for cartpole-swingup-sparse, why with IR it is still a bit worse than SAC?

**Time Spent Reviewing:**

12

---

> ### Author Response · Authors · 2021-08-10
> **Official Comment for Reviewer XWno**
>
> We thank the reviewer for their comments, which highlighted that our work was found to be significant to the community, novel, well-written and easy to follow. Below, we respond to individual questions:
>
> 1. Collapse vs. exploration: poor exploration may cause embedding collapse unless precautions are taken; in sparse reward environments, if an agent fails to explore the environment, it may never encounter a positive reward. This results in the rewards being uninformative/low-dispersion (e.g., all zeros), which leads to collapse due to the structure of the on-policy bisimulation metric. Lemma 3 characterizes this notion mathematically, showing a linear relationship between dispersion and average distance. This motivates our use of intrinsic rewards to improve exploration and thus increase robustness to collapse.
>
> 2. Our final experiments with the DeepMind Control Suite (DMC) are indeed MuJoCo environments. The cartpole-swingup-sparse environment of DMC has sparse rewards as the name suggests, and we observe the same tendency of DBC to struggle in this case. In particular, while our method reaches 200 total evaluation rewards in ~200K steps, DBC takes more than 2x as many timesteps to reach this mark. Note also that the RL agents here work with raw pixel observations (i.e., high-dimensional input) to represent observations/states. Our findings from these experiments are consistent with our findings from Fig. 2.
>
> 3. While one might argue that the slight differences between our approach and SAC in Fig. 3 (cartpole-swingup-sparse) might be due to randomness (highly overlapping shaded areas), we would like to note that SAC is a generally efficient algorithm, while DBC and our approach specifically aim to be robust/invariant to noise and distractors. As such, the DBC paper showed that DBC outperforms strong baselines under heavy distraction, but their results showed that it performed sub-par without distractions (i.e., worse than non-bisimulation-based baselines, as we also observe in the DMC tasks). Here, despite a lack of distractions, our method's performance is competitive with SAC and significantly above DBC, indicating that we remedy DBC’s apparent weakness under low-distraction conditions, thus extending its scope of applicability. We will update the manuscript to further clarify this point. We also note that, following reviewer mSYo’s recommendation, we are currently performing experiments for DMC with natural video distractions in the background. We have obtained preliminary results on the cheetah-run task, which show better results compared to both SAC and DBC. We intend to add these experiments to the final manuscript.

---

> > ### Author Response · Authors · 2021-08-23
> > **Update on Additional Experiments**
> >
> > Under point 3, we discussed the preliminary results of our additional experiments with natural video distractions. We discuss the final results of those experiments in an update to Reviewer mSYo [below](https://openreview.net/forum?id=ySFGlFjgIfN&noteId=8PnB2CRmi24).

---

### Decision · Program_Chairs · 2021-09-27

**Decision:**

Accept (Poster)

**Comment:**

Three knowledgable reviewers recommended acceptance of the paper (2x accept, 1x weak accept) and one reviewer recommended (weak) rejection of the paper. The authors addressed most of the reviewers' concerns in their rebuttal but some concerns were not resolved. In the discussion about the paper we came to the conclusion that the paper can provide several interesting insights but needs to address several of the reviewer concerns the camera ready version. I am therefore recommending acceptance of the paper and at the same time strongly advise the authors to carefully adjust the paper to address the remaining reviewers' concerns (in particular reviewer pEeB's concerns regarding clarity and theoretical statements).